# Development of a Normal Porcine Cell Line Growing in a Heme-Supplemented, Serum-Free Condition for Cultured Meat

**DOI:** 10.3390/ijms25115824

**Published:** 2024-05-27

**Authors:** Yeon Ah Seo, Min Jeong Cha, Sehyeon Park, Seungki Lee, Ye Jin Lim, Dong Woo Son, Eun Ji Lee, Pil Kim, Suhwan Chang

**Affiliations:** 1Department of Physiology, University of Ulsan College of Medicine, Ulsan 05505, Republic of Korea; dldlruddlek@naver.com (Y.A.S.); realbe0802@naver.com (M.J.C.); lyej1014@naver.com (Y.J.L.); dwson00@naver.com (D.W.S.); hyloveej6@hanmail.net (E.J.L.); 2Research Group of Novel Food Ingredients for Alternative Proteins, The Catholic University of Korea, Bucheon 14662, Republic of Korea; ntewec@catholic.ac.kr (S.P.); kimp@catholic.ac.kr (P.K.); 3Department of Biotechnology, The Catholic University of Korea, Bucheon 14662, Republic of Korea; sklee345@catholic.ac.kr; 4Asan Medical Center, Seoul 05505, Republic of Korea

**Keywords:** porcine cell, heme, cultured meat, serum free, cytochrome P450, family 1, subfamily A, polypeptide 1 (CYP1A1), glutathione peroxidase 5 (GPX5), lactoperoxidase (LPO)

## Abstract

A key element for the cost-effective development of cultured meat is a cell line culturable in serum-free conditions to reduce production costs. Heme supplementation in cultured meat mimics the original meat flavor and color. This study introduced a bacterial extract generated from *Corynebacterium* that was selected for high-heme expression by directed evolution. A normal porcine cell line, PK15, was used to apply the bacterial heme extract as a supplement. Consistent with prior research, we observed the cytotoxicity of PK15 to the heme extract at 10 mM or higher. However, after long-term exposure, PK15 adapted to tolerate up to 40 mM of heme. An RNA-seq analysis of these heme-adapted PK15 cells (PK15H) revealed a set of altered genes, mainly involved in cell proliferation, metabolism, and inflammation. We found that cytochrome P450, family 1, subfamily A, polypeptide 1 (CYP1A1), lactoperoxidase (LPO), and glutathione peroxidase 5 (GPX5) were upregulated in the PK15H heme dose dependently. When we reduced serum serially from 2% to serum free, we derived the PK15H subpopulation that was transiently maintained with 5–10 mM heme extract. Altogether, our study reports a porcine cell culturable in high-heme media that can be maintained in serum-free conditions and proposes a marker gene that plays a critical role in this adaptation process.

## 1. Introduction

Cultured meat (or meat alternatives) offers a chance to address the current threats posed by large-scale, conventional meat production [1]. Especially environmental effects and animal welfare issues are recognized as major drawbacks to the sustainability of animal farming [2]. Despite this need, the marketability of cultured meat remains low, and one of the main reasons is the high cost of the production of large cell mass in vitro [3]. A significant portion of the production cost for cultured meat is attributed to cell culture media, accounting for 55% to 90% [4], with serum price posing a significant burden [5,6]. Therefore, developing a practical and cost-effective method to culture animal cells without serum is crucial. Several serum substituents for cultured meat have been reported, ranging from *Chlorella* algae extract [7] to a Beef-R culture medium that replaces albumin with rapeseed protein isolate [8]. Another challenge hindering the consumption of cultured meat is its lack of natural meat flavor [9]. The flavor of conventional meat derives from various ingredients, including fat, organic acids, and blood. The distinctive flavor of blood comes primarily from its main component, heme (combined with iron), which is contained in red blood cells [10]. Additionally, Fe-heme incorporation can add a natural reddish color to the cultured meat. A previous study using a bovine myosatellite cell culture mentioned that extracellular heme proteins influence the color of cell-based meat [10]. However, an excess amount of heme in culture media causes cytotoxicity [11]. Therefore, the appropriate amount of heme in the cell culture needs to be determined, dependent on the cell type of interest. Moreover, it is important to obtain a resource of heme that is cheap enough to be added to cultured meat production. In this regard, heme produced from bacteria can be an attractive option along with algae, plants, or yeast. A previous study showed that heme produced from *Corynebacterium* can be used as a food additive [12]. Additionally, another study showed heme production for artificial meat in *Corynebacterium glutamicum* via systems metabolic and membrane engineering [13].

This study aimed to utilize the extract of *Corynebacterium* enriched with a high content of heme that is obtained through directed evolution [14]. Due to its lack of genetic engineering, any cultured meat utilizing this heme extract is expected to have less risk. Additionally, we expected that the bacterial extract could be used as a nutrient source in reduced-serum or serum-free conditions. Considering the safety issues of the cultured meat, we selected normal (untransformed) cells from the porcine kidney, PK15. We successfully derived a subculture from this cell line that thrives in high-heme conditions, and the molecular analysis of these cells is presented here.

## 2. Results

### 2.1. Culture of PK15 Cells under a Serial Increase of Heme Extract Concentrations Generates a Heme-Adapted Subculture

PK15 porcine kidney cells [15] are normal cells that grow well in Dulbecco’s modified Eagle’s medium (DMEM)-10% fetal bovine serum (FBS), and we found that the growth was not compromised until 2% FBS condition (Appendix A). However, this cell displayed heme toxicity at 0.4 mg/mL after 48 h (Figure 1A, blue line with triangle mark), as previously shown in other cells [11]. We also observed reduced cell proliferation by treatment with 5–10 mM of heme in PK15 cells (Figure 1B). These data imply that bacterial heme, like other sources such as reticulocytes, also induces toxicity in eukaryotic cells [11]. To overcome heme toxicity in normal cells under natural (non-genetically modified organisms) conditions, we adopted a 3T3 culture method with increasing amounts of heme extract starting from 5 μM and increasing up to 40 mM (for heme measurement, see Section 4 and Appendix A). The resulting adapted cells, named PK15H, showed comparable morphology under heme-supplemented media (5 mM) compared with the original PK15 (Figure 1C). This result indicated that the porcine cell line PK15 is adaptable to heme toxicity and raised the possibility that cells are suitable for the object of our study. 

### 2.2. RNA-seq Analysis Reveals an Increased Apoptotic Process in Heme Extract-Treated PK15 Cells

To understand the molecular changes underlying the toxicity triggered by heme treatment, RNA-seq analysis was performed in the control (PK15_F2H0) and heme extract-treated cells (PK15_F2H10). To identify heme-dependent gene expression changes, we also included treatment with original Corynebacterium extract (without directed evolution [12], marked as CCH10). The principal component analysis (PCA) plot in Figure 2A indicates each treatment group (circle for 24 h and triangle for 28 h of treatment). We could see a clear difference between F2CCH10 and F2H10 in 48 h, implying that high-heme treatment induces changes in the expression profile in PK15 cells. The volcano plot in Figure 2B reveals a range of significantly altered genes, with 74 to 314 upregulated genes and 86 to 833 downregulated genes, indicating considerable changes in gene expression following heme treatment. Top upregulated/downregulated genes are shown in Figure 2C for all combinations of comparison (See Appendix A for *p*-values). 

Among these, the expression changes of glutathione peroxidase 5 (*GPX5*), cytochrome P450, family 1, subfamily A, polypeptide 1 (*CYP1A1*), and lactoperoxidase (*LPO*) were further analyzed using real-time PCR (Figure 3A–C, Appendix A for other candidates). We selected these genes based on previous reports suggesting the function of these genes on detoxification or oxidative stress responses ([16,17,18], see discussion). Gene Ontology (GO) analysis of the RNA-seq data indicated that heme treatment upregulated the apoptotic process and innate immune responses (Appendix A). This is consistent with the biological process enrichment results shown in Figure 3E and Appendix A. We considered that bacterial components (such as lipopolysaccharides), could trigger innate immune response, while heme toxicity also transiently upregulates the apoptotic response. 

### 2.3. Heme-Adapted PK15 Cells Display a Distinct Expression Signature

Using the RNA-seq data from transient heme extract treatment as a guide, we proceeded to generate PK15 cells stably adapted under high-heme-supplemented media. The PK15H cells were obtained through subculture under increasing concentrations of heme up to 40 mM (Figure 4A). Following adaptation, these cells exhibited superior proliferation compared to the control PK15 cells under high-heme conditions (Figure 4B). Considering these cells are maintained with only 1% of FBS, it raises the possibility that PK15H cells are not only adaptable to high-heme but also to serum-free conditions. To comprehend the adaption of PK15 cells to high-heme conditions, we conducted another transcriptome analysis comparing PK15(wt) with heme-adapted PK15 cells. In this RNA-seq analysis, PK15H5, PK15H20, and PK15H40 were compared with the original PK15H0. The distribution of expression levels for PK15H0 and PK15H5/20/40 showed no drastic changes, as depicted in Appendix A. However, the PCA plot showed clear differences between the control and heme-adapted groups, and further change was observed in high-heme-tolerable cells (Appendix A, red and green dots).

Consistently, the one-way hierarchical heatmap indicated clear expression changes between the control and the heme-adapted cells (Figure 5A, Appendix A for hierarchical clustering). Therefore, we concluded that the heme adaptation process generated a subclone of PH15 cells with different expression profiles that helped these cells become viable under heme-mediated toxicity or oxidative stress. The volcano plot revealed numerous significantly altered genes (Figure 5B, Appendix A for the scatter plot), and the top differential genes summarized in Table 1. Pathway analysis revealed that several metabolic processes were highly enriched, implying that the adapted cell established a novel metabolism to endure high-heme conditions (Figure 5C, Table 2 for the top-ranked pathway list). For instance, genes encoding the NADH dehydrogenase complex show upregulation, while those associated with beta-catenin binding are downregulated in heme-adapted cells (Appendix A). These findings suggest that the heme adaptation triggers metabolic alterations linked to oxidative stress [19]. Among the various upregulated genes identified in heme-adapted cells (Table 1), we initially focused on examining the expression of CYP1A1, GPX5, and LPO, as their levels were altered in the transient-heme treatment experiment (Figure 3A–C). Real-time PCR analysis confirmed the upregulation of CYP1A1, GPX5, and LPO in heme-adapted cells (Figure 5D). Western blot analysis also supported the upregulated protein expression in heme-adapted cells (Figure 5E). These data indicated the CYP1A1, GPX5, and LPO as significant marker genes during the heme adaptation process. Additionally, we found the downregulation of thrombospondin 1 (*THBS1*) and upregulation of heme oxygenase 1 (*HMOX1*) and cytochrome P450 family 2 subfamily R member 1 (*CYP2R1*), Appendix A). However, Western blot analysis showed no difference in the THBS1 protein in response to the heme treatment (Appendix A).

### 2.4. Bacterial Extract with High-Heme Content Transiently Supports PK15 Cell Growth in Serum-Free Media

The core technology for cultured meat is to produce cell mass in serum-free conditions, as it drastically reduces the production cost. As we have seen that the PK15 cells can grow in 1% of FBS, we tried to culture the cells in serum-free conditions. In this test, we also included PK15H cells that were adapted to high-heme conditions. Control PK15 under serum-free media showed a marked change in their morphology (Figure 6A), followed by cell death within 72 h (Figure 6B, black line). When we put the PK15H cells under serum-free conditions, we found that some populations continued to grow (Figure 6A) and survived more significantly (Figure 6B, at 72 h). We found the cells are sustainable with up to 4–5 passages in serum-free conditions, but after that, it quickly lost its viability. RNA-seq analysis for the cells in serum-free conditions revealed a set of gene expression change that is distinct from either control cells or heme-adapted cells (Figure 6C), suggesting that the adaptation for serum-free conditions triggers another level of molecular changes. PCA analysis among control, heme-adapted, and serum-free cells showed a high level of separation (Figure 6D) that was further supported by hierarchical clustering analysis (Appendix A). So, these data demonstrate that serum-free condition changes PK15H cells drastically. The volcano plot in Figure 6E shows a large number of differentially expressed genes in serum-free conditions. The top altered genes are listed in Appendix A. Further analysis of the biological process and molecular function for serum-free cells indicated significantly correlated pathways, including metabolic process (Appendix A) and transcription regulator activity (Appendix A), in agreement with the expression changes shown in Figure 6C. These data suggest that PK15 cells can be transiently maintained in serum-free conditions with bacterial heme supplement. Considering the serum-free conditions are known to pose a stress that pushes cells to metabolic changes, and it is mediated by transcriptional changes molecularly [20,21], our RNA-seq results support the previous findings.

## 3. Discussion

Our study presents the use of heme extract derived from safe bacteria, *Corynebacterium*, to culture normal porcine cells. The heme-rich *Corynebacterium* has been derived by directed evolution, which does not include any artificial chemical or mutagen to ensure it is applicable to food [12]. Compared to a previous study that used the heme extract in a lactic acid bacteria culture [12], this is the first report showing the application of heme-rich bacterial extract to a eukaryotic culture system. We derived a porcine cell line adapted to high-heme culture conditions (Figure 4A,B) that normally shows toxicity (Figure 1). Overcoming heme toxicity has been rarely addressed in eukaryotic cells. Overexpression of the heme oxygenase gene in rabbit endothelial cells was shown to have a protective effect on heme and hemoglobin toxicity [22]. Another report showed that hemopexin plays a role in overcoming heme-driven oxidative stress [23]. Oxidative stress has been demonstrated to be one of the major mechanisms of heme-driven cytotoxicity [24]. Consistent with these findings, our RNA-seq data also identified several upregulated genes involved in detoxification or reactive oxygen species) homeostasis, such as CYP1A1, LPO, and GPX5 [17] (Figure 3 and Figure 5).

CYP1A1 induction has been observed in response to various stimuli such as benzo[a]pyrene (in T-47D human breast cancer cells [25]), sodium arsenide (in rats [26]), and cadmium/chromium [27]. The unexpected upregulation of CYP1A1 expression in our heme-adapted PK15 cells contradicts previous findings, as oxidative stress typically suppresses CYP1A1 expression [28]. Furthermore, CYP1A1 induction has been associated with apoptosis in several contexts [16,29]. However, a study involving CYP1A1-deficient mice suggested a protective role of the CYP1A1 against oxidative stress [30], raising the possibility that the upregulated CYP1A1 in PK15H cells serves a similar protective function. LPO is an enzyme that uses the prosthetic heme group and forms a lactoperoxidase system in bovine milk [17]. The lactoperoxidase system inactivates a wide range of microorganisms by catalyzing the (SCN^−^) by hydrogen peroxide (H_2_O_2_) to hypothiocyanite (OSCN^−^) [18,31]. In mammalian cells, the LPO was shown to play a role in the protection of hydrogen peroxide toxicity [32], and the hydrogen peroxide seems to degrade heme, according to a recent report [33]. GPX5 is known to protect oxidative stress-induced DNA mutation and lipid peroxidation [17]. GPX5 is an epididymal secretory glutathione peroxidase expressed in the mammalian male reproductive tract. Chabory et al. examined the phenotype of Gpx5-/- mice and found that the deletion of this gene did not seem to affect fertility per se [32], implying another role for this protein. A report has shown that glutathione peroxidase-1 contributes to HO-1 to redox balance in mouse brains [26], suggesting GPX5 can work similarly to help HO-1 function in PK15H cells. We propose that PK15H cells upregulate hydrogen peroxide to induce heme degradation and avoid its toxicity, which subsequently induces peroxidases to reduce the toxicity of hydrogen peroxide.

After generating PK15H cells successfully, we proceeded to drive the PH15H subpopulation sustainably in serum-free conditions (Figure 6). We observed that PK15H cells could be maintained under serum-free conditions (Figure 6A), albeit with compromised growth (Figure 6B). Serum-free adaptation of normal cells has recently been reported for immortalized chicken fibroblast cultured in suspension, providing a pathway for cultured chicken meat production [34]. Similarly, bovine progenitor cells (satellite cells) have been cultured in serum-free conditions using specific media called Beefy-9 [6]. In the case of porcine cells, LLC-PK1 (kidney epithelial) cells were adapted in serum-free media containing Prolifix, a plant-derived reagent containing the mitogenic molecule called GCR1003 [35]. We will examine this agent compared with soybean extract as a cheap and promising additive in the serum-free adaptation. With the current development scheme presented here (Figure 7), these efforts will ultimately provide us with useful information for the production of cultured pork at a reasonable cost.

## 4. Materials and Methods

### 4.1. Cell Culture and Media

HEK293 cell lines were maintained in DMEM (Hyclone, Logan, UT, USA) supplemented with 10% FBS (Hyclone, Logan, UT, USA) and 1% penicillin/streptomycin (Hyclone, Logan, UT, USA). PK15 cell lines were maintained in DMEM (Hyclone, Logan, UT, USA) supplemented with 10% FBS (Hyclone, Logan, UT, USA) and 1% penicillin/streptomycin (Hyclone, Logan, UT, USA). All cells were cultured at 37 °C in an atmosphere containing 5% CO_2_.

### 4.2. Heme and Serum-Free Adaptation Culture

PK15 cell lines [15] were purchased from KCLB (Cat No. 10033, Korean Cell Line Bank, Seoul, Korea) and cultured under the media condition of FBS 10% DMEM, gradually reduced the concentration of FBS to media of FBS 5% DMEM, FBS 2% DMEM, and FBS 1% DMEM. The culture continued with the final FBS 1% DMEM. During the subculture, the media was removed, 5 mL of PBS was added and washed, and 1 mL of 0.5% trypsin was treated and incubated in a 37 °C incubator for 3–5 min. Afterward, cells were collected using FBS 10% DMEM media and centrifuged at 25 °C 1000 rpm for 5 min. The cells collected as pellets were re-suspended with 1 mL of DMEM supplemented with 1 mL of FBS, and the pellets were released into single cells, followed by 1:3 seeding [36]. Then, heme adaptation was gradually performed with four concentrations of 5, 10, 20, and 40 µM. It concentration was increased after stable cell growth was observed for at least two passages at each concentration. At the final stage of the adaptation, the passage number of PK15H was 16 for 40 mM. The serum-free adaptation of cells that thrived under media conditions with 1% FBS and 5 µM heme extract was subjected to media with FBS 0% and 5 µM heme extract. Simultaneously, seeding was performed with 1% of FBS and a heme extract of 5 µM for the attachment of the cell, and media change was performed with 0% of FBS and a heme extract of 5 µM the next day.

### 4.3. Cell Proliferation Assay

Cell viability was assessed using Quantimax (BioMax, Gyeonggi-do, Republic of Korea), as per the manufacturer’s protocol [37]. Cells were seeded at a density of 4 × 10^3^ cells/200 μL in each well of a 96-well plate and incubated at 37 °C for 24, 48, and 72 h. To monitor cell proliferation, 1/10 volume of Quantimax was added after each time point. The cells were incubated until reaching an appropriate OD value for each day. The absorbance of the wells was measured at 450 nm using a Molecular Devices microplate reader (Winooski, VT, USA).

### 4.4. Heme Sonication and Quantitation

First, 10 g of dried heme overexpressed *corynebacterium* [12] and 40 mL of autoclaved double-distilled water (DDW) was mixed in a 50-mL tube, placed in the sonicator with the pulse setting with 100% Amp1 and 2 s of pulse plus 5 s rest. After 20 min of sonication, the tube was rested for 10 min on ice, followed by an additional 10 min of sonication under the same conditions. After centrifuging for 20 min at 4 °C and 4000 rpm, 1 mL of the supernatant was distributed in a 1.5 mL microtube and centrifuged for 15 min at 4 °C 14,000 rpm. A clear supernatant was picked and transferred to the bottle, 5% glycerol was calculated and autoclaved. It was used by aliquoting 1 mL each in a clean bench and refrigerated in a box at 4 °C to reduce exposure to light. The concentration (unit: µM) was measured using the heme assay kit (Sigma, Heme assay kit, MAK316, St. Louis, MO, USA). The concentration was measured on a half-scale in compliance with the protocol of the heme assay kit. In addition, the concentration of total protein was also quantified using a BCA assay of heme extract. The concentration of total protein corresponding to the specific concentration of heme was calculated as 52.3 mM per mg/mL of bacterial extract.

### 4.5. Protein Extraction and Western Blotting

Cells were lysed in RIPA buffer (Biosesang, Gyeonggi-do, Republic of Korea) containing protease (Roche, Mannheim, Germany) and phosphatase inhibitors (Roche, Mannheim, Germany). Lysates were centrifuged at 13,000 rpm for 15 min, and the supernatant was collected. Protein quantification was performed using BCA (PierceTM BCA Protein Assay Kit, Thermo Fisher Scientific, Rockford, IL, USA). Subsequently, the proteins were separated via sodium dodecyl sulfate-polyacrylamide gel electrophoresis and transferred to a nitrocellulose membrane (Thermo Fisher Scientific, Rockford, IL, USA). Immunoblotting was performed with antibodies, CYP1A1 (sc-101828, Santa-Cruz Biotechnology, Dallas, Texas, USA), THBS1 (sc-59887, Santa-Cruz Biotechnology, Dallas, Texas, USA), GPX5 (18731-1-AP, Proteintech, Rosemont, IL, USA), LPO (MBS9611882, MyBioSource, San Diego, CA, USA), glyceraldehyde-3-phosphate dehydrogenase (K200057M, SolarBio, Beijing, China), and β-actin (sc-47778, Santa-Cruz Biotechnology, Dallas, Texas, USA).

### 4.6. RNA Preparation and RT-PCR

RNA extraction from cells was carried out using a Tri-RNA reagent (Invitrogen, Carlsbad, CA, USA). RNA was used for cDNA synthesis (PrimeScript RT reagent kit, TaKaRa, San Jose, CA, USA). Real-time PCR was conducted with SYBR Green (AMPIGENE^®^ qPCR Green Mix Lo-ROX, Enzo Life Sciences, Farmingdale, NY, USA). The mixture of SYBR Green, cDNA, and primers was placed in Hard-Shell 96-Well PCR Plates (BIO-RAD, Hercules, CA, USA). Then, qPCR was performed in the CFX Connect Optics Module (BIO-RAD, Hercules, CA, USA). Gene expression was normalized using the reference gene actin as an internal control. The sequences for primers used in this study are listed in the Appendix A.

### 4.7. Statistics

GraphPad Prism 5.0 software (Cambridge Intelligence, Boston, MA, USA) was used, and statistical analysis was performed using an unpaired two-tailed Student’s *t*-test. The data are presented as the mean ± SEM of triplicate measurements in a representative experiment. Values of *p* < 0.05 were considered significant. Values of *p* < 0.05, *p* < 0.01, and *p* < 0.001 are designated using *, **, and ***, respectively.

## Figures and Tables

**Figure 1 ijms-25-05824-f001:**
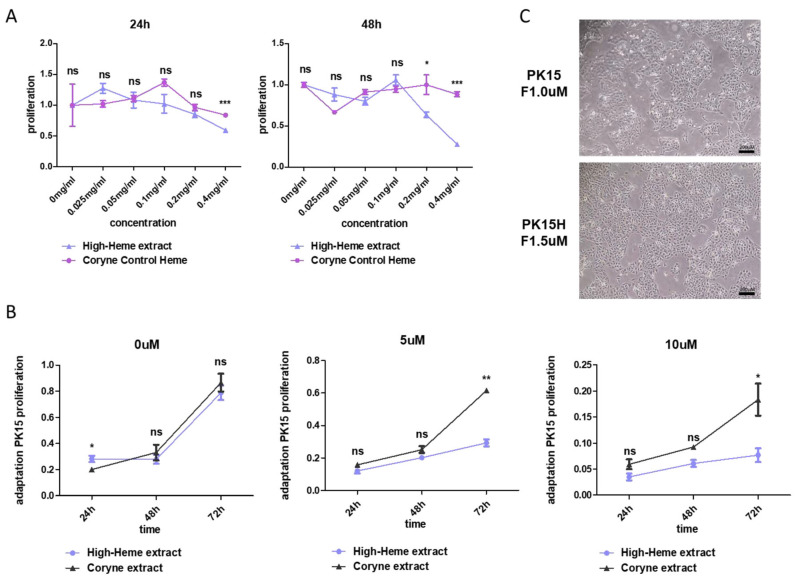
Normal porcine PK15 cells exhibit heme cytotoxicity, which can be overcome by adaptation culture. (**A**) Relative proliferation of PK15 cells under various amounts of bacterial heme extract supplemented in 24 or 48 h. The red line indicates dose-dependent proliferation upon control *Corynebacterium* extract treatment, and the blue line for high-heme-containing *Corynebacterium* extract. (**B**) Time-dependent heme toxicity in PK 15 cells. The black line indicates proliferation with control extract, whereas the blue line with high-heme extract, ranging from 0 to 10 mM. ns; not significant, * *p* < 0.05, ** *p* < 0.005, *** *p* < 0.001. (**C**) Pictures of PK15 cells under control media (without heme extract, upper panel) and cells with 5 mM of heme extract-containing media (lower panel). Scale bar: 200 mM.

**Figure 2 ijms-25-05824-f002:**
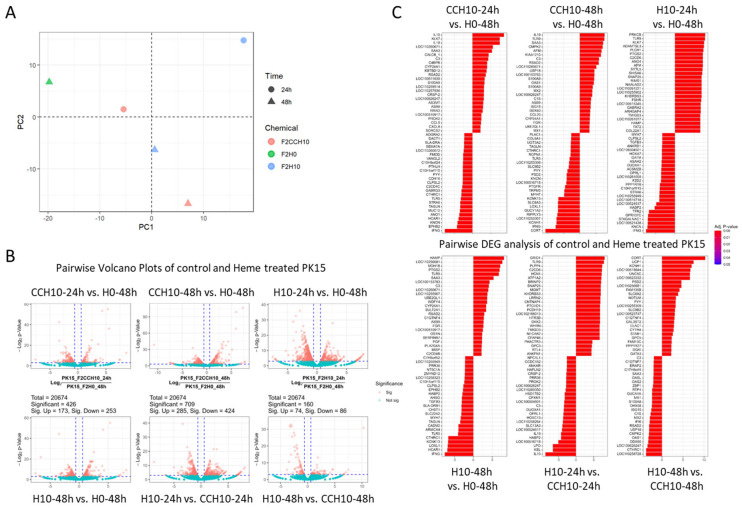
RNA-seq analysis shows dynamic molecular alteration in PK15 cells after the treatment with heme (**A**) PCA plot illustrates the expression difference among the control extract (F2CCH10, in orange), heme extract-treated (F2H10, in blue), and normal media-fed (F2H0, in green, 48 h only) PK 15 cells. Expression was monitored at 24 h (circle) or 48 h (triangle). (**B**) Volcano plots display pairwise gene expression changes for the samples mentioned in (**A**). The numbers of significantly altered (up or downregulated, in orange spots) genes are indicated at the top of each graph. (**C**) List of genes significantly up or downregulated in the pairwise gene expression analysis described in (**B**).

**Figure 3 ijms-25-05824-f003:**
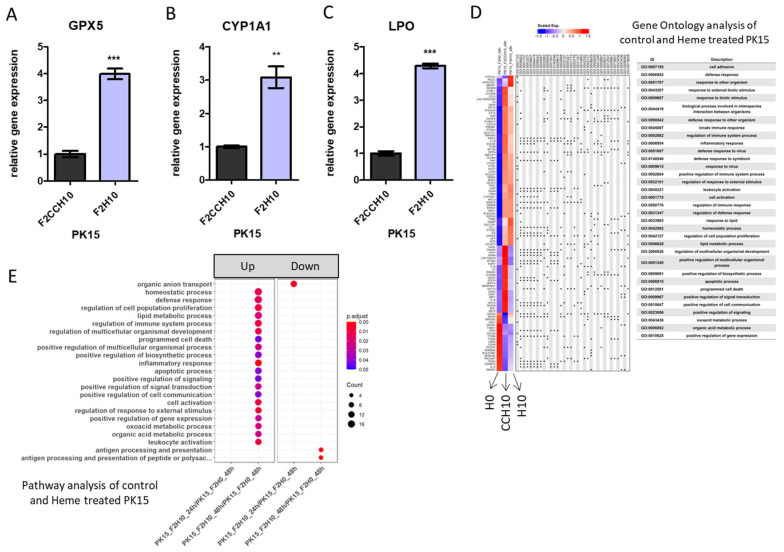
Pathway analysis for the heme extract-treated PK15 cells reveals an upregulated apoptotic pathway and suppressed immune response with several novel target genes. (**A**–**C**). Real-time PCR analysis of *GPX5* (**C**), *CYP1A1* (**D**), and *LPO* (**E**) for F2CCH10 (control) and F2H10 (heme-treated). ** *p* < 0.01, *** *p* < 0.001. (**D**). Summary of the GO clustering analysis of the expression profile of F2H0, F2CCH10, and F2H10. The color indicates upregulated (in red) or downregulated (in blue) expression. Gene names are marked on the left end, while the identity of gene ontology in which each gene is involved is listed in a table format. (**E**) Biological function analysis for the differentially expressed genes between F2H0 (control) and F2H10 (heme-treated). The size of the circle indicates the counts of each function affected, and the color shows the adjusted *p*-value turning to red for a lower *p*-value.

**Figure 4 ijms-25-05824-f004:**
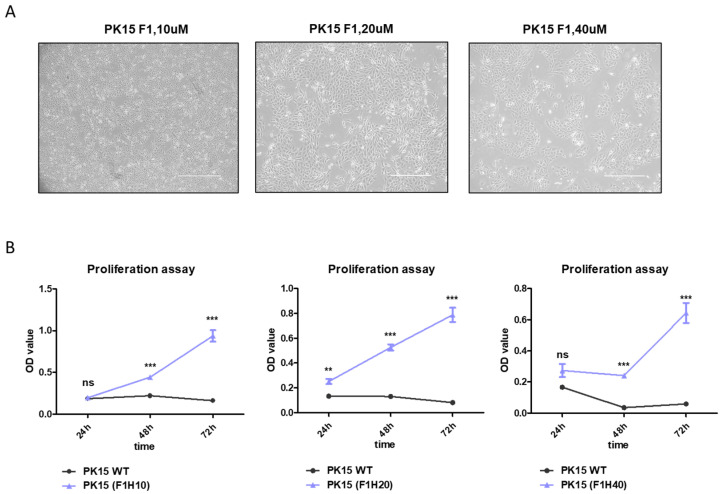
Generation of heme-adapted PK15H cells shows distinct proliferation potential up to 10 mM of heme-extract treatment. (**A**) Representative images of PK15H cells adapted in 10 mM (left), 20 mM (middle), and 40 mM (right) of heme extract-supplemented media (scale bar: 400 mM). (**B**) Graphs illustrating the proliferation of PK15H cells during 24–72 h of culture period. The black line indicates WT PK15 cells, while the blue line depicts PK15H cells proliferating under 10 mM (left), 20 mM (middle), and 40 mM (right) of heme-containing media. ** *p* < 0.01, *** *p* < 0.001.

**Figure 5 ijms-25-05824-f005:**
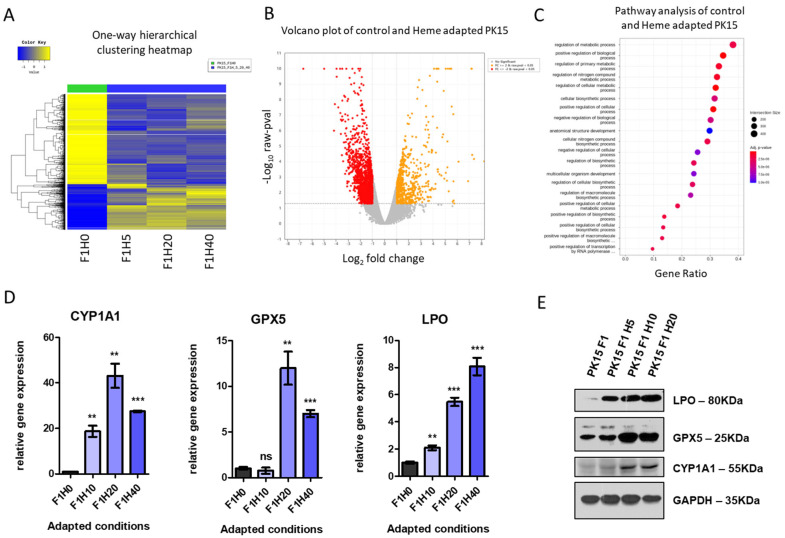
Heme-adapted PK15H cells show a distinct expression signature with altered biological pathways. (**A**) One-way hierarchical clustering heat map for the expression profiles of PK15H cells. Control (PK15-F1H0) was compared with heme-adapted cells under 10 mM (F1H10), 20 mM (F1H20), and 40 mM (F1H40). (**B**) Volcano plot showing the distribution of genes with altered expression in PK15H cells (10–40 mM). Orange dots indicate significantly upregulated genes, while red dots indicate downregulated genes. (**C**) Representative data of GO analysis showing the top 20 biological processes altered in PK15H cells (10–40 mM). The size of each circle indicates the intersection size, and the color shows the adjusted *p*-value. (**D**) Real-time PCR analysis of *GPX5* (**C**), *CYP1A1* (**D**), and *LPO* (**E**) PK15H cells adapted in 10–40 mM of heme. ** *p* < 0.01, *** *p* < 0.001. (**E**) Western blotting results for the LPO, GPX5, and CYP1A1 in PK15H cells cultured under 5–20 mM of heme. Glyceraldehyde-3-phosphate dehydrogenase was shown as a loading control.

**Figure 6 ijms-25-05824-f006:**
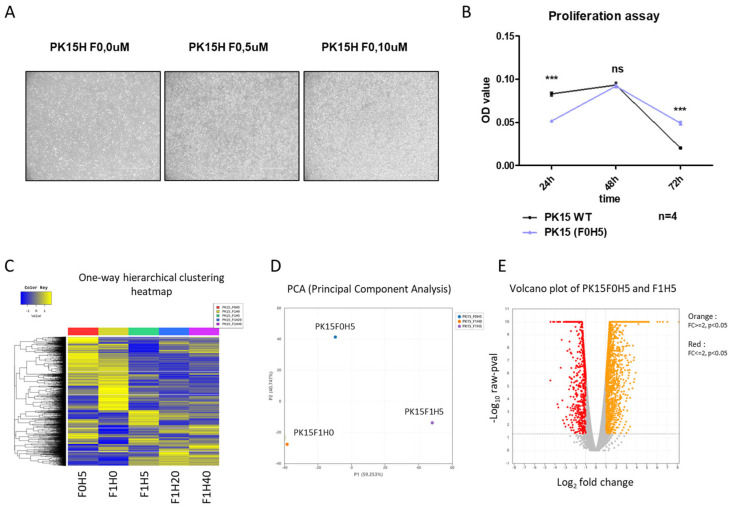
Heme-adapted PK15H cells proliferate up to five passages in serum-free media supplemented with bacterial heme extract (**A**) Representative pictures of heme-adapted PK15H cells under serum-free conditions supplemented with 5 mM (middle) and 10 mM (right). Scale bar; 400 mM. Note a superior number of proliferating cells in the presence of heme extract. (**B**) Proliferation assay of PK15WT (black line) compared with PK15H (blue line) under serum-free conditions. n = 4, ns; not significant, *** *p* < 0.001. (**C**) One-way hierarchical clustering heat map of the PK15H cells under serum-free (F0H5, red bar at the top) compared with high-heme conditions. Note the distinct expression pattern of the F0H5 sample. (**D**) PCA analysis of expression of PK15 cells under serum-free (F0H5, in blue), compared with normal (F1H0, in orange) or heme-adapted (F1H5, in purple). (**E**) Volcano plot showing differentially expressed genes under serum-free conditions supplemented with heme extract. Orange dots indicate upregulated genes, while red dots show significantly downregulated genes.

**Figure 7 ijms-25-05824-f007:**
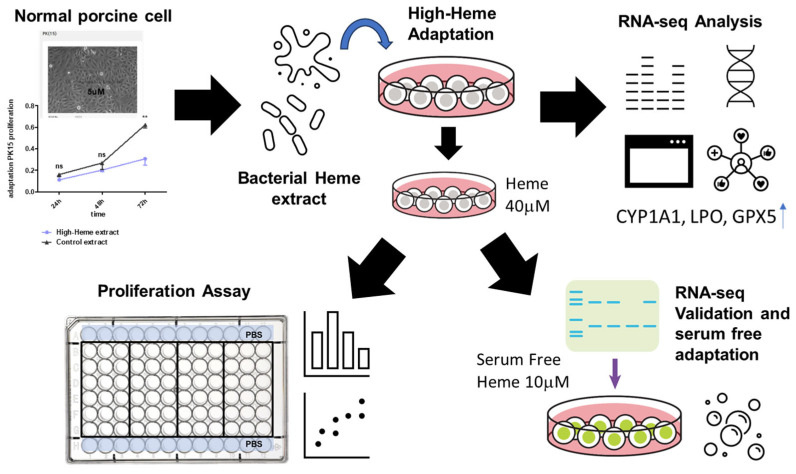
Summary of the study with the development scheme of PK15H.

**Table 1 ijms-25-05824-t001:** List of top genes differentially expressed in PK15 cells after high-heme adaptation.

Gene_ID	Transcript_ID	Gene_Symbol	Description	PK15_F1H_5_20_40/PK15_F1H0.fc	PK15_F1H_5_20_40/PK15_F1H0.logCPM	PK15_F1H_5_20_40/PK15_F1H0.raw.pval	PK15_F1H_5_20_40/PK15_F1H0.bh.pval
100049691	NM_001098597	OSTN	osteocrin	142.744964	6.229014	4.9843 × 10^−33^	7.5307 × 10^−29^
100520218	XM_003132298,XM_013981851,XM_013981853,XM_013981854,XM_013981855,XM_021070779,XM_021070780,XM_021070781	ADAMTS9	ADAM metallopeptidase with thrombospondin type 1 motif 9, transcript variant X1	−104.994789	1.934609	1.2212 × 10^−28^	9.2253 × 10^−25^
100152394	NM_001244640,NM_001244641	RTN1	reticulon 1	20.525415	5.414605	3.2805 × 10^−20^	1.6522 × 10^−16^
100513560	XM_013979738,XM_013979739,XM_013979740,XM_013979741,XM_021063401,XM_021063402,XM_021063403	ETV1	ETS variant 1, transcript variant X7	16.908247	6.160216	1.4798 × 10^−19^	5.5895 × 10^−16^
100737483	XM_021091686	LOC100737483	keratin, type II cytoskeletal 6A	45.172367	5.278455	3.7805 × 10^−18^	1.1424 × 10^−14^
403103	NM_214412,XM_021098317	CYP1A1	cytochrome P450 1A1	21.822831	4.754509	1.0019 × 10^−17^	2.523 × 10^−14^
110260665	XM_021091687	LOC110260665	keratin, type II cytoskeletal 6A-like	32.843664	6.004388	4.9299 × 10^−17^	1.0641 × 10^−13^
100624115	XM_021100590	ADGRL3	adhesion G protein-coupled receptor L3	−6.342352	5.136530	1.3642 × 10^−16^	2.5765 × 10^−13^
106507259	XR_002343949	LOC106507259	keratin, type II cytoskeletal 6A-like	41.605454	4.962795	1.7087 × 10^−16^	2.8685 × 10^−13^
102164968	XM_013984758,XM_021091688,XM_021091689	KRT6A	keratin 6A, transcript variant X2	44.892932	8.266441	4.7843 × 10^−16^	7.2286 × 10^−13^

**Table 2 ijms-25-05824-t002:** List of the top pathways altered in PK15 cells adapted in high-heme conditions.

Term_id	Term_Name	Adjusted_*p*-Value	Term_ Size	Query_Size	Intersec-tion_Size	Effective_Domain_Size	Intersection_Size_UP	Intersection_Size_ DOWN
GO *:0005737	cytoplasm	1.15186 × 10^−10^	9056	1107	628	19,669	288	340
GO:0031323	regulation of the cellular metabolic process	5.11208 × 10^−8^	4452	1060	339	19,136	111	228
GO:0048522	positive regulation of the cellular process	5.11208 × 10^−8^	4306	1060	330	19,136	123	207
GO:0048518	positive regulation of the biological process	5.19773 × 10^−8^	4924	1060	367	19,136	144	223
GO:0051171	regulation of nitrogen compound metabolic process	9.9293 × 10^−7^	4665	1060	344	19,136	120	224
GO:0080090	regulation of the primary metabolic process	1.10779 × 10^−6^	4793	1060	351	19,136	122	229
GO:0009891	positive regulation of the biosynthetic process	1.53459 × 10^−6^	1613	1060	146	19,136	43	103
GO:0031325	positive regulation of the cellular metabolic process	2.19895 × 10^−6^	2373	1060	196	19,136	63	133
GO:0010557	positive regulation of macromolecule biosynthetic process	2.19895 × 10^−6^	1520	1060	138	19,136	39	99
GO:0031328	positive regulation of cellular biosynthetic process	2.19895 × 10^−6^	1580	1060	142	19,136	41	101

* GO, Gene Ontology.

## Data Availability

The RNA-seq data are available upon request if they will be used for academic purposes only.

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
