# Peer review of "Development of a Normal Porcine Cell Line Growing in a Heme-Supplemented, Serum-Free Condition for Cultured Meat"

_ijms, 2024, doi:10.3390/ijms25115824_

Round 1
Reviewer 1 Report
Comments and Suggestions for Authors
The present article has some innovation; however, I have some concerns.
First of all, the article needs to be improved. The English is not clear. The idea of this article needs to be strongly ameliorated.
The discussion needs to be ameliorated, compared with other studies, as well as the conclusion.
What concentrations do you used of Corynebacterium extract? Why do you obtained them and perform their extracts?
Cite other articles, discuss the novelty of your work, please, sell your idea.
In addition:
Change microM to µM
Revise all English, including space
Corynebacterium
Mentioned cell passage and origin …
“Hrs” no, “h”
Figure 1C, needs to be ameliorated

Comments on the Quality of English LanguageMajor English revisions are needed.
Author Response
Point by point responses to reviewers comments
Reviewer #1
The present article has some innovation; however, I have some concerns.
First of all, the article needs to be improved. The English is not clear. The idea of this article needs to be strongly ameliorated.
>> We appreciate the critical comment and regret that our English was not clear to deliver our idea. We first thoroughly revised our manuscript to improve it by ourselves and then send it to professional English Editing Service to correct and finalize it. Please check the final version as well as certificate for the English Editing Service.
The discussion needs to be ameliorated, compared with other studies, as well as the conclusion. Cite other articles, discuss the novelty of your work, please, sell your idea.
>> We are very thankful for the valuable comment and agree that Discussion part was not enough to describe the novelty and impact of our study. Following the suggestion, we revised the Discussion part to add related references and describe the novelty of our study. Also, we added a schematic presentation at the end of Discussion, to summarize our study.
What concentrations do you used of Corynebacterium extract? Why do you obtained them and perform their extracts?
>> We think this is a critical question. We obtained it from our collaborator who have developed high-heme producing Corynebacterium and applied it to animal food. We added corresponding reference in the Introduction and Discussion section.
In addition:
Change microM to µM >> We changed all micro to µ
Revise all English, including space >> We checked all space use in the manuscript
Corynebacterium >> We italicized it
Mentioned cell passage and origin …>> We added the origin of PK15 (source) and mentioned passages of its derivatives, PK15H, in methods section (At the final stage of the adaptation, the passage number of PK15H was 16 for 40mM Heme).
“Hrs” no, “h” >> We changed all hrs to h
Figure 1C, needs to be ameliorated
>> We agree that the image of Figure 1C was poor. Following the comment, we revised and placed enlarged image of the cell picture with new scale bar.

Reviewer 2 Report
Comments and Suggestions for Authors
Report on the manuscript ijms-2994726 entitled: Development of normal porcine cell line growing in heme-supplemented, serum free condition for cultured meat.
The authors conclude L.320-325:
All these efforts will open a promising way to obtain cultured meat with reasonable price, and our study add another way to achieve this using a bacterial extract enriched with heme. Although the serum free adaptation of PK15H is not complete as it sustains only 5 passages, we hope to improve this by optimizing culture condition or adding cheap supplement such bean extracts. Further study will help us to find answer for these possibilities.
However, no such conclusions can be deduced or inferred from the results.
A mere explanation of the caption of each table and figure is stated in the “Results” section. An in-depth description of the results shown is mandatory.
The “Discussion” section considers mostly general issues. In fact, it is completely detached from both the “Results” section and the aim/scope of the paper. It needs to be rewritten.
To be reviewed:
Figure 1a and b, Figure 3a, b, and c, Figure 4d, Figure 5d, and figure 6b:
The SD bars should show symmetry to the mean value.
Result and Discussion sections should be improved accordingly.
Comments on the Quality of English Language--
Author Response
Point by point responses to reviewers comments
Reviewer #2
The authors conclude L.320-325:
All these efforts will open a promising way to obtain cultured meat with reasonable price, and our study add another way to achieve this using a bacterial extract enriched with heme. Although the serum free adaptation of PK15H is not complete as it sustains only 5 passages, we hope to improve this by optimizing culture condition or adding cheap supplement such bean extracts. Further study will help us to find answer for these possibilities.
However, no such conclusions can be deduced or inferred from the results.
>> We thank for the critical comment and agree that this part is over-stated without supporting data. Therefore, we deleted this paragraph. Also, we edited the last part of our abstract correspondingly.
A mere explanation of the caption of each table and figure is stated in the “Results” section. An in-depth description of the results shown is mandatory.
>> We regret that the result section merely describe the data without in-depth interpretation. Following the comment, we revised Result section throughly and added in-depth description for most of the results.
The “Discussion” section considers mostly general issues. In fact, it is completely detached from both the “Results” section and the aim/scope of the paper. It needs to be rewritten.
>> We thank to the helpful comment and agree that the Discussion section did not contain aim/scope of this study and impact of the results. We therefore entirely revised discussion section and added such contents. Also, we added a schematic presentation of this study at the end of discussion to summarize our results.
To be reviewed:
Figure 1a and b, Figure 3a, b, and c, Figure 4d, Figure 5d, and figure 6b:
The SD bars should show symmetry to the mean value.
>> We really appreciate for the helpful comment. We revised all the graphs mentioned and set new SD bars.
Result and Discussion sections should be improved accordingly.
>> We regret that the result section merely describe the data without in-depth interpretation. Following the comment, we revised Result section thoroughly and added in-depth description for most of the results. Also, we entirely revised discussion section and added aim/scope of this study and impact of the results. Lastly, we added a schematic presentation of this study at the end of discussion to summarize our results.

Reviewer 3 Report
Comments and Suggestions for Authors
This research article by Seo et al. aims to use the extract of Corynebacterium enriched with a high content of heme (obtained by direction evolution) for reducing the risk levels in cultured meat. This is an interesting piece of work carried out methodically with some novelty and scientific quality. However, the following issues need to be addressed before this paper could be accepted for publication:
1. The unit mention “microM” should be corrected by replacing “micro” with its symbol throughout the manuscript.
2. The unit consistency should be carefully double-checked throughout the manuscript. For example, minutes and hours should be abbreviated as “min” and “h” as well as the word “degree” should be replaced with its symbol. Also, “ml” should be “mL” and “u” should be replaced with “micro” symbol.
3. There should be a space between the value and unit, as well as between the word and reference citations throughout the manuscript.
4. All the purchase details of chemicals/reagents and instruments/equipment/software should be provided as state, city, and country in the case of USA as well as city and country in the case of other countries. Also, the authors can just mention the company name for the second instance.
5. A reference citation should be included for the procedures provided under sections 2.3, 2.4, 2.5, and 2.6.
6. The clarity of all the figures is VERY poor and needs to be enlarged for clarity/redrawn. I am wondering how the authors themselves were able to read the figure contents.
7. At the end of discussion, a schematic diagram should be included to depict the overall mechanism.
8. All the abbreviations should be provided in full form in the first instance with the abbreviation in parenthesis and only in abbreviated form in subsequent places.
9. All the abbreviations used in both Tables and Figures should be provided in full form in the respective Table footnotes and Figure captions.

Comments on the Quality of English LanguageMinor editing of English language required
Author Response
Point by point responses to reviewers comments
This research article by Seo et al. aims to use the extract of Corynebacterium enriched with a high content of heme (obtained by direction evolution) for reducing the risk levels in cultured meat. This is an interesting piece of work carried out methodically with some novelty and scientific quality. However, the following issues need to be addressed before this paper could be accepted for publication:
- The unit mention “microM” should be corrected by replacing “micro” with its symbol throughout the manuscript.
>> We thank for the helpful comment and regret that we missed it. Now we corrected all of the micro to symbol m.
- The unit consistency should be carefully double-checked throughout the manuscript. For example, minutes and hours should be abbreviated as “min” and “h” as well as the word “degree” should be replaced with its symbol. Also, “ml” should be “mL” and “u” should be replaced with “micro” symbol.
>> We appreciate for the helpful comment and regret that unit was not consistent in our manuscript. Now we corrected all the mentioned units including min, h, °C, mL and m.
- There should be a space between the value and unit, as well as between the word and reference citations throughout the manuscript.
>> We thank for the detailed comment. We revised proper space throughout the manuscript.
- All the purchase details of chemicals/reagents and instruments/equipment/software should be provided as state, city, and country in the case of USA as well as city and country in the case of other countries. Also, the authors can just mention the company name for the second instance.
>> We appreciate for the valuable point. Following the comment, we revised the method section and added manufacture’s information.
- A reference citation should be included for the procedures provided under sections 2.3, 2.4, 2.5, and 2.6.
>> We appreciate for the helpful comment. We revised the method section and added manufacture’s information accordingly.
- The clarity of all the figures is VERY poor and needs to be enlarged for clarity/redrawn. I am wondering how the authors themselves were able to read the figure contents.
>> We regret that the resolution of the figure is very poor. We revised all figures with small marks to enlarge it make it sure its quality. We really thank for the helpful comment that improved readability of Figures. Please also check the Main Figure file provided separately.
- At the end of discussion, a schematic diagram should be included to depict the overall mechanism.
>> We appreciate for the valuable comment and added one schematic diagram at the end of discussion section (Provided as Figure 7).
- All the abbreviations should be provided in full form in the first instance with the abbreviation in parenthesis and only in abbreviated form in subsequent places.
>> We are very grateful for the detailed point. Following the comment, we carefully revised the abbreviations and provided its full name on its first appearance.
- All the abbreviations used in both Tables and Figures should be provided in full form in the respective Table footnotes and Figure captions.
>> We appreciate for the detailed point. Following the comment, we carefully revised the abbreviations in Figures and Tables and provided its full name on its first appearance.

Round 2
Reviewer 1 Report
Comments and Suggestions for Authors
Dear Authors, thanks for improving the paper.